:Ọ: PLOS | ONE

# ART is key to clearing oncogenic HPV genotypes (HR-HPV) in anal mucosa of HIV-positive MSM

Carmen Hidalgo-Tenorio[1]*, Concepción Gil-Anguita[1], Miguel Angel López Ruz[1], Mohamed Omar[2], Javier López-Hidalgo[3], Juan Pasquau[1]

1 Department of Infectious Diseases; "Virgen de las Nieves" University Hospital, Granada, Spain, 2 Unit of Infectious Diseases, Jaen "Hospital Complex", Spain, 3 Pathology Department; "Virgen de las Nieves" University Hospital, Granada, Spain

* chidalgo72@gmail.com

**Data Availability Statement:** All relevant data are within the manuscript and its Supporting Information files.

**Funding:** The author(s) received no specific funding for this work.

## Abstract

### Background

Anal squamous cell carcinoma (ASCC) is one of the most frequent non-AIDS-defining neoplasias in HIV patients, mainly in MSM, and it has been associated with chronic infection with high-risk human papilloma virus (HR-HPV). Our main objective was to determine HR-HPV clearance and acquisition rates and related factors and their relationship with the incidence of HSILs and ASCC in anal mucosa of HIV+ MSM.

### Patients and methods

The study included consecutive HIV-infected MSM between May 2010 and December 2018. Data were gathered at baseline and annually on their sexual behavior, CD4 and CD8 levels, plasma HIV viral load, and results of anal cytology, HPV PCR, and high-resolution anoscopy.

### Results

Out of the 405 patients studied, 34.9% of patients cleared oncogenic genotypes (IQR: 37–69) within 49 months, and 42.9% acquired new genotypes within 36 months (IQR:12–60). In multivariate analysis, clearance was only significantly influenced by the duration of antiretroviral therapy (ART) (OR: 1.016, 95% CI 1.003–1.030). The incidence of HSILs was 30.86/1,000 patient-years and that of ASCC was 81.22/100,000 patient-years; these incidences were not influenced by the acquisition (acquired: 14.9% *vs.* non-acquired: 10.4%; p = 0.238) or clearance (cleared 11.4% *vs.* non-cleared: 13.2%; p = 0.662) rates of these viruses.

### Conclusions

The duration of ART appears to positively affect oncogenic genotype clearance in the anal mucosa of HIV+ MSM, although this clearance does not affect the incidence of HSILs or ASCC. The reduction in HSIL+ rate observed in our patients may be attributable to the bundle of measures adopted at our center.

**Competing interests:** The authors have declared that no competing interests exist.

## Introduction

Anal squamous-cell carcinoma (ASCC) is rare in the general population, with an incidence of around 2 per 100,000 people/year [1]. However, it is one of the most frequent non-AIDS-defining neoplasias among HIV-infected patients, especially in men who have sex with men (MSM), with a high incidence of up to 131 per 100,000 people/year [1] and a similar survival rate to that in the general population [2]. ASCC has been associated with persistent infection by high-risk oncogenic HPV (HR-HPV) genotypes, which increases the expression of onco-proteins E6 and E7, activates the cell cycle, inhibits apoptosis, and allows the accumulation of damaged DNA [3]. HIV and HPV share similar risk factors and interact in various ways to infect the anal-genital area; their interaction has been reported to double the risk of HPV acquisition and halve the clearance rate among seropositive individuals in comparison to the general population [4].

Preventive measures against HPV acquisition include circumcision in males [5], use of condom [6], and HPV vaccination [7]. Utilization of the condom is deficient among MSM [8], especially since the availability of pre-exposure prophylaxis to reduce the risk of HIV transmission [9]. The resulting relaxation in precautions taken by HIV+MSM has led to changes in sexual practices, including a greater use of "chemsex" drugs [10] and an increase in outbreaks of sexually transmitted infection [11] and in HIV diagnoses reported in Spain [12] and other developed countries [13]. The HPV vaccine has not proven effective against associated high-grade squamous epithelial lesions (HSILs), although it appears to reduce their recurrence [14]. In addition, the ACTG trial on the quadrivalent HPV vaccine in seropositive adults was suspended for lack of efficacy [15]. Finally, many HIV+MSM are already HIV-infected, and they are generally over the maximum age (26 years) recommended for vaccination [16].

Given the lack of clarity on factors associated with HPV clearance in anal mucosa of HIV+MSM patients [4] and the consequent absence of specific recommendations, the main objectives of this study were to determine the clearance rate of oncogenic HPV genotypes and associated factors and to explore their possible relationship with the incidence of HSIL and ASCC in the anal mucosa of a prospectively enrolled cohort of HIV+MSM. Further objectives were to study: risk factors for HPV infection; genotype distribution as a function of histology; and correlations among cytology, HPV polymerase chain reaction (PCR), and histological findings in the anal mucosa.

## Patients and methods

### Design

We conducted a longitudinal, prospective, single-center study of 405 HIV+ MSM patients consecutively enrolled between May 2010 and December 2018 in a program for the screening, diagnosis, treatment, and follow-up of dysplastic anal mucosa lesions conducted by the Infectious Disease Department of our hospital. The inclusion criterion was being an HIV+ MSM aged >18 years with no history of ASCC.

At their baseline visit (V0), we obtained the written informed consent of the patients to participation in the study, which was approved by the ethical committee of our hospital. Information was then gathered on their epidemiological, clinical, and analytical data. All data were treated in accordance with Spanish data protection legislation Law 15/1999, 13 December, on Personal Character Data Protection).

At V0, data were collected on clinical-epidemiological variables, including: age, history of anal-genital warts, current anal-genital warts, total number of sexual partners to date, number of sexual partners during the past 12 months, age at first sexual intercourse, frequency of condom

utilization, tobacco smoking (yes/no; packets/year), alcohol consumption (yes/no; standard drink units [SDUs]/day), injecting drug use (IDU), Ex-IDU, nationality, educational level, months since HIV diagnosis, HIV stage (CDC classification), months on antiretroviral therapy (ART), the antiretrovirals received, virological failure (defined by viral-RNA >50 copies/mL in ≥ 2 measurements during previous 6 months), and concomitant treatment(s). Information was also recorded on the presence/absence of: chronic liver disease due to hepatitis B (HBV) or C (VHC) virus; positive luetic serology; other sexually transmitted disease(s); and latent, treated, or active tuberculosis infection. Analytical data were also gathered on CD4 nadir, CD4 lymphocyte counts, and viral load at HIV diagnosis and at V0. In addition, PCR (*Gonorrhoeae*, *Mycoplasma spp*, *Chlamydia spp*, *Ureaplasma spp*) and oral-anal-urethral exudate culture (*Gonorrhoeae*) studies were performed in patients with symptomatology or infected partner.

At V0 and all subsequent visits, cotton swabs soaked in saline solution were used to take two anal canal mucosa samples: one for HPV detection and genotyping by qualitative PCR (Linear Array HPV Genotyping Test) using a "GeneAmp PCR System 9700" thermocycler (Applied Biosystems, Roche, Switzerland); and the other for cytology study using the "thin-layer" technique (Procesador Thin Prep 2000 (Hologic). Both samples were immersed in thin-layer liquid medium and sent to the pathology laboratory of the hospital. Genotypes 16, 18, 26, 31, 33, 35, 39, 45, 51–53, 56, 58, 59, 66, 68, 73, and 82 were considered to be HR-HPV genotypes. Genotypes 6, 11, 34, 40, 42–44, 54, 55, 57, 61, 70–72, 81, 83, 84, and 89 were considered to be low-risk (LR-HPV) genotypes. Genotypes 39, 45, 59, and 68 were classified as subtypes of HPV genotype 18; and genotypes 31, 33, 35, 52, 58, and 67 as subtypes of HPV 16 [17].

At the next visit, 4–12 weeks later, high-resolution anoscopy (HRA) was performed with a Carl Zeiss 150 FC colposcope (Carl Zeiss, Oberkochen, Germany). After an initial rectal examination, a transparent disposable anoscope was introduced and used to instill 5 mL acetic acid, which was left in place for 3 min before removal of the anoscope and inspection of the mucosa. The anoscope was then used to instill 5% Lugol's iodine, followed by further inspection of the mucosa. Samples were taken, from all four quadrants, of apparently normal mucosa and of areas with Lugol-negative aceto-white lesions. At the same visit, biopsies of the anal mucosa were taken using an endoscopic retrograde cholangiopancreatography catheter.

Follow-up visits: The timing of the next follow-up visit depended on the results of the above studies. Patients with normal anoscopy and LSIL (AIN1) were evaluated one year later with cytology, HPV PCR, and anoscopy studies. There were two options for patients with HSIL: mucosectomy with electric scalpel by the coloproctology unit of the General Surgery Department (from May 2010 onwards); or self-administration of 5% imiquimod 3 times weekly for 16 weeks (from November 2013 onwards). Once their treatment was completed, these patients underwent another anoscopy, and those with normal histology or LSIL were given an appointment one year later, while those with HSIL were re-treated. When ASCC was detected, patients were referred to the oncology department for treatment. Variables gathered at the annual follow-up visits were the same as at V0.

Between May 2012 and May 2014, the quadrivalent HPV vaccine was administered to 66 patients with no HPV 16 or 18 infection and no previous or current lesions compatible with HSIL+.

Cytology results were categorized in four types in accordance with the Bethesda classification [18] as: atypical squamous cell (ASC), atypical squamous cell-high grade (ASC-H), low-grade squamous intraepithelial lesion (LSIL), and high-grade squamous intraepithelial lesion (HSIL).

Histological results were classified in accordance with the Lower Anogenital Squamous Terminology (LAST) standardization project for HPV as: LSIL (AIN1/condyloma), HSIL (AIN2, AIN3), or anal squamous cell carcinoma (ASCC) [19].

## Definition of variables

**HR-HPV clearance.** The disappearance of all HR-HPV genotypes from anal canal mucosa in the latest examination, with negative HPV PCR

**HR-HPV acquisition.** The acquisition of new oncogenic HPV genotypes in anal mucosa, detected by HPV PCR.

**Abnormal cytology.** Cytology result of ASCUS, LSIL, or HSIL.

**Histology with anal HSIL+.** Anal mucosa biopsy result ranging from high-grade squamous intraepithelial lesion to anal cancer.

**Histology with LSIL+.** Anal mucosa biopsy result ranging from low-grade squamous intraepithelial lesion to anal cancer.

## Statistical analysis

A descriptive analysis was performed, calculating central tendency and dispersion measurements (means, standard deviations, medians, percentiles) for quantitative variables and absolute frequencies with 95% confidence interval (CI) for qualitative variables. We calculated the prevalence of HPV, anal cytology, and histology findings with 95% CI interval. Factors potentially related to oncogenic genotype clearance and prevalence were analyzed in bivariate analyses. The Student's t-test for independent samples was applied for quantitative variables with a normal distribution and the Mann-Whitney U test for those with a non-normal distribution. Qualitative variables were analyzed with Pearson's chi-square test or, when application criteria were not met, with Fisher's exact test. The Kolmogorov-Smirnov test was used to check the normality of variable distribution. The degree of concordance between HR-HPV PCR and biopsy results was determined with the Kappa index [20].

Multivariate logistic regression analyses were conducted using Freeman's formula [n = 10* (k+1)] [21]. The model for factors related to oncogenic virus clearance included variables that were significant in bivariate analyses (ART duration, consumption of non-analogs, and infection with low-risk HPV genotypes) as well as the number of sexual partners during previous 12 months and during the follow-up, smoking habit, age, and use of condom. The model for factors related to the prevalence of oncogenic HPV genotypes included history of genital warts, CD4 nadir levels, infection with low-risk HPV genotype, CD4 level, HIV viral load, CD4/CD8 ratio, undetectable viral load, receipt of ART, consumption of analogs and integrase inhibitors, age (over vs. under 50 years), smoking, sexual behaviors, genital warts at time of anoscopy, time since HIV diagnosis, and ART duration. Variables were selected for models using a stepwise approach, with significance levels of 0.05 for entry and 0.10 for exit. The Hosmer-Lemeshow test was used to analyze the goodness of fit of the models.

A significance level of 0.05 was considered for all tests. SPSS 21.0 was used for the statistical analysis.

## Results

### 1. Baseline characteristics of the cohort

We consecutively included 405 patients with a mean age of 36 years, 56.7% were university students, 52.8% were smokers, and they had a mean of 1 (IQR 1–7) sexual partner over the previous 12 months. The median time since HIV diagnosis was 2 years; mean CD4 nadir was 367.9 cells/uL, 86.7% were receiving ART, mean CD4 level was 689.6 cells/uL, and mean CD4/CD8 ratio was 0.77; 85.9% of patients were undetectable (Table 1).

Among the 394 participants, 76.9% were infected with a median of 1 high-risk genotype (IQR: 1–3), 73.1% with a median of 1 low-risk genotype (IQR: 0–2), and 58.1% by both low-

**Table 1. Characteristics of HIV+MSM patients.**

|  | Number of patients (n = 405) |
|---|---|
| **Age** (± SD) | 36.2 (± 10.1) |
| <30 yrs, n (%) | 123 (30.4) |
| 30–50 yrs, n (%) | 244 (60.2) |
| >50 yrs n (%) | 38 (9.4) |
| **Retired, n (%), 95% CI** | 6 (1.5) |
| **Educational level** |  |
| No studies | 40 (9.9) |
| Primary studies | 129 (31.9) |
| Secondary studies | 230 (56.7) |
| University studies | 22 (5.4) (3–7.5) |
| **Origin** |  |
| Europe | 387 (95.6) |
| Central America | 17 (4.2) |
| **qHPV vaccine (2012–2014), n (%), 95% CI** | 66 (16.3) |
| **Age at first sexual intercourse, median (IQR)** | 18 (16–20) |
| **Number of lifetime male sex partners, median (IQR)** | 50 (15–150) |
| **Number of male sex partners during previous 12 months, (IQR)** | 1(1–7) |
| **Habitual use of condoms,** n (%), 95%CI | 294 (72.6) (68.2–77.4) |
| **Total number of sexual partners during follow-up, median (P25-P75)** | 54.5(20–154) |
| **History of anal/genital warts,** n (%), (95%CI) | 128 (31.6), (27.1–36.1) |
| **Anal/genital warts at baseline,** n (%), (95%CI) | 93 (23), (18.6–26.8) |
| **History of Syphilis,** n (%), IC95% | 103 (25.4), (21.6–29.8) |
| **History of other STI,** n (%), IC 95% | 110 (27.2), (23.1–31.6) |
| **Time since HIV diagnosis (months), median** (IQR) | 25 (8–84) |
| **CD4 at diagnosis of HIV (cel/uL),** (± SD) | 448± 298.17 |
| **HIV viral load at diagnosis of HIV (log), mean (IQR)** | 4.61 (4.07–5.12) |
| **CD4 nadir (cells/uL),** (± SD) | 367.93±233.85 |
| **CD4 nadir < 200 cells/uL, n(%), 95% CI** | 97 (23.9), (20–28.5) |
| **CD4 cell count at baseline (cells/uL),** (± SD) | 689.64± 475.03 |
| **CD8 cell count at baseline (cells/uL),** (± SD) | 981.5±531.5 |
| **CD4 /CD8 ratio, mean, (± SD)** | 0.77±0.70 |
| **HIV viral load at baseline (log), median, IQR** | 0 (0–1.72) |
| **Undetectable: < 50 HIV RNA copies/mL of plasma n (%)** | 348 (85.9) |
| **History of AIDS diagnosis,** n (%), 95% CI | 106 (26.2) (21.3–30.1) |
| **HAART before inclusion, n (%), 95% CI** | 351 (86.7), (83.2–90) |
| **Previous ART line, mean (IQR)** | 1 (1–2) |
| **Virological failure, n (%)** | 17 (4.8) |
| **Median months of ART,** (IQR) | 4 (16–56) |
| **Chronic HCV infection,** n (%) | 14 (3.5) |
| **Chronic HBV infection,** n (%) | 13 (3.2) |
| **Smoker, pack/year, mean (IQR)** <br> **Smoker, n (%), 95% CI** | 1.5 (0–14) <br> 214 (52.8) (47.9–57.4) |
| Alcohol, SDU, mean, (IQR) | 0 (0–4) |
| **EX-IDU,** n (%) | 2 (0.5) |

HCV, **chronic infection by hepatitis C virus**; HBV, **chronic infection by hepatitis B virus**; EX-IDU, **ex-injecting drug users**; STI: **sexual transmitted infection;** VL**: viral load; SDU: standard drink unit;** IQR: **interquartile range;** SD, **(standard deviation)**

and high-risk genotypes. The most frequent genotype was genotype 16 (in 27.7%), followed by low-risk genotypes 6, 11, and 42 (each in 18%). Results for remaining variables are exhibited in Table 2.

In the baseline cytology study, 47.9% of patients had LSIL, 41.6% normal cytology, 7.3% atypical squamous cells of undetermined significance (ASCUS), and 3.3% HSIL. According to the high-resolution anoscopy study, 46.7% of samples were normal, 40.4% LSIL, 12.3% HSIL, and 0.5% ASCC (Table 3).

## 2. Clearance and acquisition of HR-HPV infection and prevalence of HSIL + in anal mucosa

The 405 patients were followed for a median of 36.45 months (IQR: 12–69); 301 patients (74.3%) had at least two valid HPV genotype identification studies, and 105 (34.9%) of these evidenced oncogenic virus clearance within 49 months (IQR: 37–69) and 129 (42.9%) acquired new genotypes, with a median of 1(IQR: 0–2), within 36 months (IQR: 12–60). At the end of the follow-up, 68.8% had low-grade genotypes, 59.5% had high-grade genotypes, and 32.3% had both low- and high-grade genotypes. We found no significant difference in infection by low-risk genotypes between baseline (V0) and final (Vf) visits (V0: 73.1% *vs*. Vf: 68.8%, p = 0.254); however, there were differences in the rate of infection by oncogenic viruses [V0: 76.6% *vs*. Vf: 59.5%; p = 0.0001] and in the median number of viruses [V0: 1 (IQR: 1–3) *vs*. Vf: 1 (IQR: 0–2), p = 0.0001]. We also found a reduction in simultaneous infection by low- and high-risk viruses (V0: 57.3% *vs*. Vf: 32.3%; p = 0.0001).

Among the 405 patients, 87.2% had at least two control anoscopies with a median follow-up of 36 months (IQR: 12–69). During this period, 88 HSIL cases and 3 ASCC cases were diagnosed, with prevalence rates of 21.7% and 0.74%, respectively. There were 38 new cases of HSIL (incidence of 30.86/1,000 patient-years) and 1 new case of ASCC (incidence of 81.22/100,000 patient-years); 28.9% (11/38) progressed to HSIL from normal anoscopy and 71% (27/38) from LSIL; 49 HSILs were treated by mucosectomy and 34 with intra-anal 5% imiquimod. None of the treated patients progressed to ASCC. We found no differences in HSIL+ rate between baseline and the end of the 36-month follow up [12.8% (52/405) *vs*. 12.3% (39/317); p = 0.323] or as a function of the acquisition (acquired: 14.9% *vs*. non-acquired: 10.4%; p = 0.238), or clearance (cleared: 11.4% *vs*. non-cleared: 13.2%; p = 0.662) of oncogenic genotypes. However, as depicted in Fig 1, we found significant reductions in HSIL+ cases between 2010 and 2018 (42.9% *vs*. 4.1%, p = 0.034), between 2010 and 2013 (42.9% *vs*. 13.8%, p = 0.003), and between 2013 and 2016 (13.8% *vs*. 4.8%, p = 0.0001), followed by a stabilization between 2016 and 2018 (4.8% *vs*. 4.1%, p = 0.617).

One ASCC was resolved with wide local surgery; another was treated with abdominal-pelvic amputation, chemotherapy, and radiotherapy and has been in remission for 12 months; and the third patient with ASCC died at 15 months post-diagnosis after chemotherapy and radiotherapy with a complete local response and subsequent cervical lymph-node metastasis, in which HPV genotype 33 was isolated.

There were four deaths during the follow-up: one patient with hepatic cirrhosis secondary to chronic HCV infection in 2012, one with Burkitt lymphoma in 2013, one with small-cell lung cancer in 2014, and one with metastatic ASCC in 2015.

## 3. Factors associated with clearance of oncogenic HPV genotypes

In the multivariate analysis, ART duration of >5 years was the sole factor favoring oncogenic HPV genotype clearance (OR: 1.016, 95% CI 1.003–1.030); and the number of sexual partners

**Table 2. Prevalence of HPV in anal mucosa.**

| n (%), 95% CI | HPV PCR in anal mucosa (n = 394) |
|---|---|
| HR-HPV | 303 (76.9), (73–81.1) |
| LR-HPV | 288 (73.1), (69–77) |
| HR and LR-HPV | 229 (58.1), (53–63) |
| Median HR-HPV, IQR | 1 (1–3) |
| Median LR-HPV, IQR | 1 (0–2) |
| HPV 6 | 71 (18) |
| HPV 11 | 71 (18 |
| HPV 12 | 1 (0.3) |
| HPV 16 | 109 (27.7) |
| HPV 18 | 51 (12.9) |
| HPV 26 | 6 (1.5) |
| HPV 31 | 55 (14) |
| HPV 33 | 29 (7.4) |
| HPV 35 | 36 (9.1) |
| HPV 39 | 46 (11.7) |
| HPV-40 | 7 (1.8) |
| HPV 42 | 72 (18.3) |
| HPV-43 | 10 (2.5 |
| HPV 45 | 50 (12.7) |
| HPV 48 | 1 (0.3) |
| HPV 51 | 55 (14 |
| HPV 52 | 50 (12.7) |
| HPV 53 | 36 (9.1) |
| HPV 54 | 26 (6.6) |
| HPV 55 | 64 (16.2) |
| HPV 56 | 31 (7.9) |
| HPV 58 | 23 (5.8) |
| HPV 59 | 42 (10.7) |
| HPV 61 | 30 (7.6) |
| HPV 62 | 56 (14.2) |
| HPV 64 | 1 (0.3 |
| HPV 66 | 34 (8.6) |
| HPV 68 | 42 (10.7) |
| HPV 69 | 14 (3.6) |
| HPV 70 | 32 (8.1) |
| HPV 71 | 1 (0.3) |
| HPV 72 | 28 (7.1) |
| HPV 73 | 37 (9.4) |
| HPV 81 | 51 (12.9) |
| HPV 82 | 17 (4.3) |
| HPV 83 | 5 (1.3) |
| HPV 84 | 30 (7.6) |
| HPV 89 | 1 (0.3) |
| HPV 6108 | 13 (3.3) |
| HPV-AR subtype of HPV 18 (39,45,59,68) | 164 (41.6) |
| HPV-AR subtype of HPV 16 (31,33,35,52,58,67) | 207 (52.5) |

HPV, **human papillomavirus;** HR-H**: high-risk HPV,** LR-HPV**: low-risk HPV**

**Table 3. Results of cytology and anoscopy for cohort inclusion.**

|  | Cohort of MSM-HIV patients (n = 405) |
| --- | --- |
| **Anal cytology,** (n = 397), n (%), 95% CI |  |
| LSIL | 190 (47.9), (43–52.7) |
| HSIL | 13 (3.3), (1.8–5.1) |
| ASCUS | 29 (7.3), (5.1–10.2) |
| Normal | 165 (41.6), (37–46) |
| **Anoscopy: Histology** (n = 405), n (%), 95% CI |  |
| Normal | 189 (46.7), (44.5–54) |
| LSIL | 164 (40.4), (36.6–46) |
| HSIL | 50 (12.3), (9.2–15.9) |
| ASCC | 2 (0.5), (0–0.8) |

LSIL, low-grade squamous intraepithelial lesion; HSIL, high-grade squamous intraepithelial lesion; ASC, atypical squamous cells of undetermined significance; ASCC, anal squamous cell cancer.

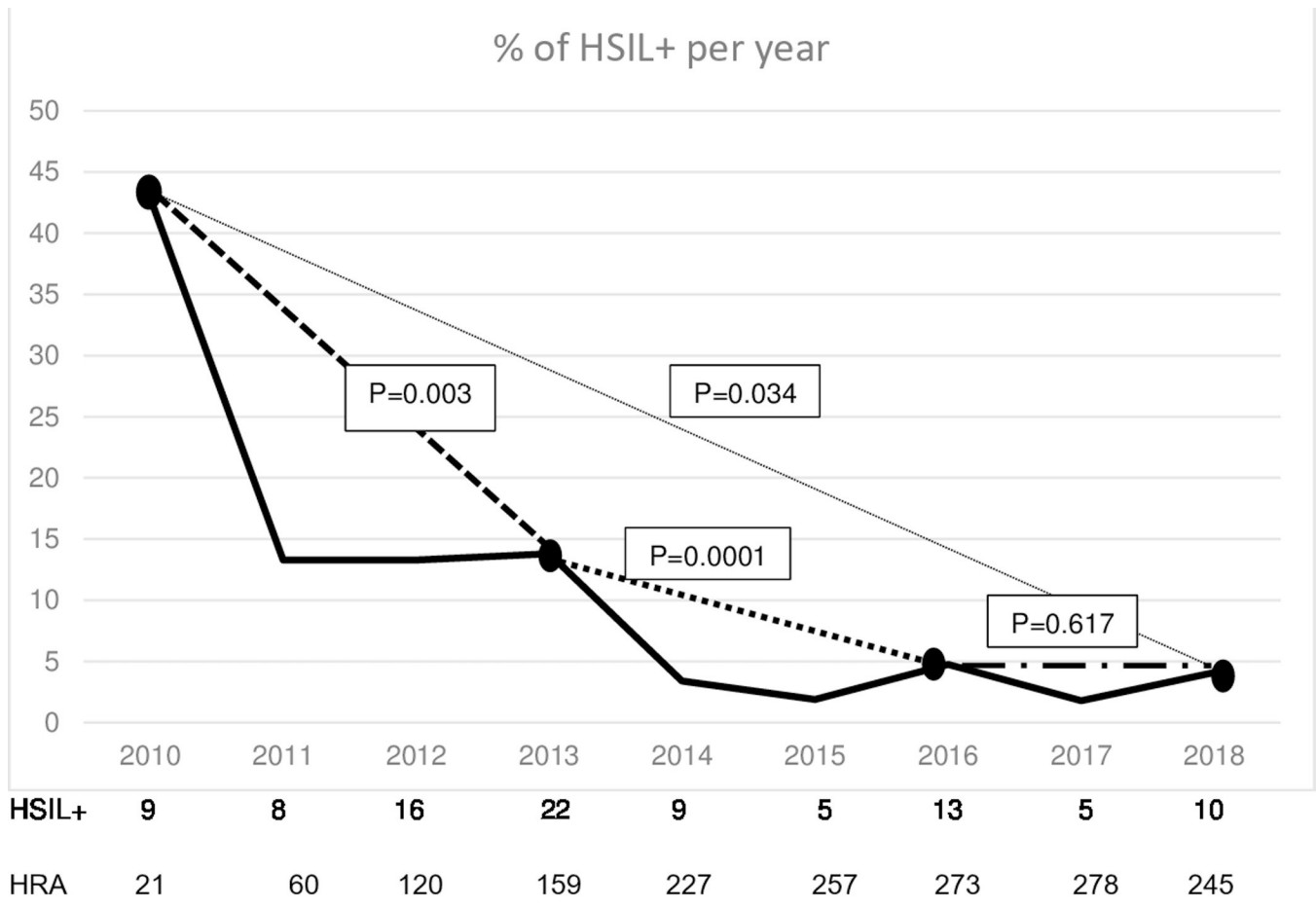

HRA: high resolution anoscopy.

**Fig 1. HSIL + rate during the following of HIV+HSH cohort.**

during the previous 12 months was the sole risk factor for persistence of oncogenic HPV genotype [1(1–3) *vs.* 1(1–8), (OR: 0.954; 95% CI: 0.911–0.998)], (Table 4).

## 4. Factors associated with infection by oncogenic HPV genotypes

In the multivariate analysis, anal mucosa infection by high-risk HPV was related to: CD4 nadir < 200 cells/uL (27.1% *vs.* 13.2%; OR 2.363, 95% CI: 1.062–5.256); and simultaneous infection by low-risk and high-risk genotypes (75.2% *vs.* 65.9%; OR 1.842, 95% CI: 1.015–3.344). Protective factors against this infection were ART with integrase inhibitors (26.7% *vs.* 42.9%; OR: 0.508, 95% CI: 0.277–0.935) (Table 5).

## 5. HPV genotype distribution by histology

Genotype 16 was the most frequently isolated oncogenic virus in all types of mucosa and was associated with the presence of LSIL+ [55.9% (67/109) *vs.* 46.5% (132/284); OR 1.837, 95% CI: 1.170–2.883; p = 0.008] but not with the presence of HSIL+ [16.5% (18/109) *vs.* 10.9% (31/284); OR: 1.614, 95% CI: 0.861–3.026; p = 0.135]

As depicted in Fig 2A, the most frequently isolated genotypes were 6 (15.4%), 16 (21.6%), 42 (18.5%), 45 (23.2%), and 62 (31.9%); none of the 36 genotypes isolated in normal mucosa was significantly associated with this finding in the histological study.

In LSIL+ mucosa samples, the most frequently isolated genotypes were 6 (19.1%), 11 (19.8%), 16 (32.3%), 31 (16.7%), 42 (19.1%), and 55 (17.3%), as shown in Fig 2B. Genotype 66 was significantly associated with LSIL (12.9% *vs.* 5.6% p = 0.001).

In mucosa samples with HSIL or ASCC (Fig 2C), the most frequently isolated HPV genotypes were 6 (25%), 11(27%), 16 (37%), 18 (22%), 39 (18%), 51 (20%), 53 (22%), 61 (19), and 68 (25). Genotypes significantly associated with the presence of HSIL were 18 (22.4% *vs.* 11.5%; p = 0.034), 53 (22.4% *vs.* 7.2%, p = 0.001), 61 (18.4% *vs.* 6.1%, p = 0.006), and 68 (24.5% *vs.* 8.7%, p = 0.001).

## 6. Correlation of anal cytology and HPV PCR with histological findings

The cytology study showed that 165 samples were normal and 232 had some degree of dysplasia.

In patients with normal cytology results, anoscopy showed normal mucosa in 36.6% (p = 0.0001), LSIL in 50% (p = 0.0001) and HSIL in 17.2% (p = 0.001).

Out of the 190 samples with LSIL by cytology, histological results by anoscopy were LSIL in 52.1% (p = 0.0001) and 16.8% were HSIL in 16.8% (p = 0.015). Thirteen samples were HSIL by cytology, representing 92.8% of all HSIL cases diagnosed by anal mucosa biopsy (p = 0.0001).

Among the 303 patients infected with oncogenic HPV genotypes, anoscopy results showed that 45.2% had LSIL (p = 0.003) and 13.8% HSIL (p = 0.115).

Notably, none of the 48 (12.1%) patients with normal cytology and negative high-risk HPV PCR result had a histological finding that was compatible with HSIL (p = 0.005) or ASCC (p = 0.1) (Table 6).

## Discussion

This study prospectively enrolled HIV+ MSM patients in a program to screen, diagnose, treat, and follow up anal mucosa dysplastic lesions. The mean age was below 40 years, with less than 10% were aged over 50 years. The majority of the patients had university studies, smoked, and were studying or employed. They reported a median of 55 sexual partners in their lives to date, and more than 70% habitually used condoms. Their viro-immunological status was excellent,

**Table 4. Risk factors associated with HR-HPV clearance in HIV+ MSM patients.** Bivariate and multivariate analysis.

| | Clearance (n = 105) | Non-clearance (n = 174) | Bivariate p* | Multivariate OR**; 95% CI |
|---|---|---|---|---|
| **Mean age (yrs)**, (± DS) | 34.83(± 10.13) | 34.48(± 9.09) | 0.812 | 0.774; (0.421–1.421) |
| <30 yrs, n (%) | 32 (30.5) | 51 (29.3) | 0.837 | |
| 30–50 yrs, n (%) | 60 (57.1) | 111(63.8) | 0.269 | |
| >50 yrs, n (%) | 13 (12.4) | 13 (7.5) | 0.172 | |
| **Retired, n (%)** | 5 (4.8) | 12 (6.9) | 0.47 | |
| **Smoker, n(%)** | 46(43.8) | 84(48.3) | 0.469 | |
| **Charlson Index, (IQR)** | 0(0–0) | 0(0–3) | 0.178 | 0.983; (0.557–1.735) |
| **qHPV Vaccine, n (%)** | 26 (24.8) | 32(18.4) | 0.2 | |
| **Age at first sexual intercourse, (IQR)** | 18 (16–21) | 18(16–20) | 0.592 | |
| **Genital/anal warts, n (%)** | 14(13.3) | 100(0–100) | 0.719 | |
| **History of Syphilis** or Other STI, n (%) | 47(44.8) | 68(39.1) | 0.35 | |
| **Total NPS, baseline visit,** (IQR) | 30(10–122) | 31(10–110) | 0.095 | |
| **NPS previous 12 mo baseline visit** (IQR) | 2(1–7) | 1(1–5) | 0.378 | |
| **Total NSP last visit,** (IQR) | 40(14.5–175) | 60.5 (21–152) | 0.113 | 1; (1–1.001) |
| **NSP12m before last visit** IQR | 1 (1–3) | 1 (1–8) | 0.096 | **0.954; (0.911–0.998)** |
| **Use of condom during study, n (%)** | 66 (62.9) | 107 (61.5) | 0.650 | 1.472; (0.749–2.892) |
| **Rate of condom use, follow-up,** (IQR) | 100 (10–100) | 100(30–100) | 0.373 | |
| **History of AIDS (A3, B3, C), n (%)** | 28(26.7) | 56(32.2) | 0.330 | |
| **Time since HIV diagnosis** (months),(IQR) | 30(11–84) | 31(10–110.5) | 0.386 | |
| **CD4 at HIV diagnosis** (cells/ul), (± SD) | 501.4(± 275.9) | 444.5(±298.4) | 0.209 | |
| **CD8 at HIV diagnosis** (cells/ul), (± SD) | 1204.7(± 1106.8) | 1105.1(± 615.4) | 0.453 | |
| **Viral load HIV at diagnosis** (log), (± SD) | 5.45(± 6.081) | 5.68(± 6.19) | 0.361 | |
| **CD4 nadir** (cells/ul), (± SD) | 349.03(±193.2) | 357.04(±255.89) | 0.784 | |
| **CD4 nadir < 200 cells/uL**, n(%) | 24(22.9) | 51(29.3) | 0.246 | |
| **CD4 nadir <500 cells/uL**, n(%) | 84 (80) | 127(72.9) | 0.164 | |
| **CD4 nadir >500 cells/uL**, n(%) | 19 (18.1) | 45 (43.3) | 0.139 | |
| **Baseline visit (inclusion)** | | | | |
| CD4 (cells/uL), (± SD) | 671.86(± 307.61) | 652.46(± 283.8) | 0.672 | |
| CD8 (cells/uL), (± SD) | 1051.44(± 459.97) | 1134.98(± 621.3) | 0.340 | |
| CD4/CD8, (± SD) | 0.73(± 0.41) | 0.689(± 0.42) | 0.54 | |
| HIV VL(log) (± SD) | 3.74(± 4.38) | 3.79(± 4.34) | 0.882 | |
| VL<50 cop/mL | 75 (71.49) | 115 (66.1) | 0.382 | |
| **Final visit** | | | | |
| CD4 (cells/uL), (± SD) | 838.72(± 295.96) | 861.15(± 1087.4) | 0.867 | |
| CD8 (cells/uL), (± SD) | 900.94(± 361.3) | 996.57(± 492.4) | 0.168 | |
| CD4/CD8, (± SD) | 0.96(± 0.38) | 0.92(± 0.82) | 0.664 | |
| HIV VL (log) (± SD) | 0.73(± 1.19) | 3.57(±4.47) | 0.295 | |
| **ART during follow-up, n (%)** | 104(99.1) | 166(95.4) | 0.095 | |
| NRTI | 96(91.4) | 160(91.9) | 0.877 | |
| NNNRTI | 64(60.9) | 79(45.4) | **0.012** | 1.53; (0.836–2.774) |
| PI | 43(40.9) | 65(37.3) | 0.55 | |
| INSRTI | 40(38.1) | 85(48.8) | 0.08 | 1.049; (0.573–1.919) |
| Median of months of ART, (IQR) | 64(25–112) | 58(27–112) | **0.003** | **1.016; (1.003–1.030)** |
| Virological failure, n (%) | 5(4.8) | 14(8) | 0.278 | |
| **Infection for Low-risk HPV genotype** | 62(59.1) | 124(71.3) | **0.025** | 0.757; (0.407–1.407) |

(*Continued*)

**Table 4.** (Continued)

| | Clearance (n = 105) | Non-clearance (n = 174) | Bivariate p* | Multivariate OR**; 95% CI |
|---|---|---|---|---|
| **Previous HSIL, n(%)** | 26(24.8) | 45(25.8) | 0.8 | |
| **Imiquimod for HSIL** | 10(9,5) | 20(11.4) | 0.61 | |
| **Surgery for HSIL** | 16(15.4) | 23(13.2) | 0.637 | |
| **Imiquimod for HSIL or warts** | 15(14.3) | 33(18.9) | 0.316 | |

P*: p-value.

OR**: crude odds ratio.

**95% CI: 95% confidence interval.** HIV+MSM, **HIV-positive men that have sex with men;** LTI, **Latent tuberculosis infection;** HCV **hepatitis C virus;** HBV, **hepatitis B virus;** HPV, **Human papillomavirus;** EX-IDU, **ex-injecting drug addict;** VL, **viral load.** HR-HPV: **high-risk HPV,** LR-HPV: **low-risk HPV;** LSIL, **low-grade squamous intraepithelial lesion;** HSIL, **high-grade squamous intraepithelial lesion;** ASC, **atypical squamous cells of undetermined significance,** NSPt, **Total number of sexual partners; NSP12m: number of sexual partners during past 12 months**

85.6% were on ART, and the mean time since HIV diagnosis was around two years. HPV infection prevalence was high, 76.9% had high-risk genotypes 73.1% had low-risk and 58.1% both types of genotype. The most frequently isolated genotype was HPV-16, followed by three low-risk genotypes (6, 11, and 42). This high prevalence of anal HPV infection has previously been reported in HIV+ MSM [22] and has been significantly associated with the onset of LSIL/ HSIL [23].

The clearance rate of oncogenic genotypes was lower than the acquisition rate. Over one-third of patients eliminated the infection within a median of 4 years; consequently, there was a lower rate of patients infected by oncogenic viruses at the end of follow-up. The duration of ART was the sole factor favoring oncogenic virus clearance (OR 1.016; 95% CI [1.003–1.030]), which was inversely related to the number of sexual partners during the previous 12 months (OR 0.954; 95% CI 0.911–0.988). A recent meta-analysis [4] concluded that a CD4 lymphocyte count ≤200 cells/uL favored the acquisition of oncogenic genotypes and hindered their clearance, and that HIV *per se* was a predictor of HPV infection.

Neither HPV clearance or acquisition rates were associated with the prevalence or incidence of HSIL. However, we have observed a significant reduction in the HSIL+ rate among HIV+ MSM since the start of the program in 2010, with a stabilization since 2016. No patients with treated HSIL progressed to ASCC. This encouraging trend may be attributable to the bundle of measures implemented at our center, including screening for anal cancer and precursor lesions and treatment with surgery and/or imiquimod, besides administration of the tetravalent VPH vaccine in around 15% of the patients. The positive effects of participation in a clinical trial should also be taken into account. A recent study of 592 HIV patients, with a mean follow-up of 69 months, reported that the risk of progression from HSIL (AIN3) to ASCC was high and that ASCC screening was the only factor that reduced this risk [24]. Publication of data from the Study for the Prevention of Anal Cancer (SPANC) [25] is expected to elucidate the natural evolution of HPV infection and allow a more effective classification of patients at risk of ASCC. In the meantime, implementation of a program to screen, diagnose, treat, and follow up anal mucosal dysplastic lesions appears recommendable, especially in HIV + MSM.

Factors related to infection with HPV oncogenic virus were a history of CD4 nadir < 200 cells/uL (OR: 2.363, 95% CI: 1.062–5.256), and simultaneous infection by low- and high-risk HPV genotypes (OR: 1.842; 95% CI: 1.015–3.344). The sole protective factor against infection was ART that included integrase inhibitors (OR: 0.508, 95% CI: 0.277–0.935). These data confirm recent observations of the protective effects of ART and CD4 levels>500 cells/uL against

**Table 5. Risk factors associated with HR-HPV genotype infection in HIV+ MSM patients. Bivariate and multivariate analysis.**

| | With HPV-AR N = 303 | Without HPV-AR N = 91 | Bivariate p[*] | Multivariate OR[**]; 95% CI |
|---|---|---|---|---|
| **Mean age (yrs)**, (± DS) | 35.94(± 9.86) | 37.04(± 11.07) | 0.362 | 1.117; (0.634–1.98) |
| <30 yrs, n (%) | 94(31) | 27(29.7) | 0.806 | |
| 30–50 yrs, n (%) | 183(60.4) | 52(57.1) | 0.579 | |
| >50 yrs, n (%) | 26(8.6) | 12(13.2) | 0.19 | |
| **Retired, n (%)** | 14(4.6) | 8(8.8) | 0.129 | |
| **Educational level**, n(%) | | | | |
| No studies | 4(1.3) | 2(2.2) | | |
| Primary studies | 28(9.2) | 10(10.9) | 0.539 | |
| Secondary-PT studies | 92(30.4) | 33(36.3) | | |
| University studies | 178(58.7) | 46(50.5) | | |
| **Smoking** | 160(52.9) | 47(51.6) | 0.846 | |
| **Smoking, (pack/yr), IQR** | 1.50(0–13.25) | 1.9(0–15) | 0.738 | 0.95; (0.971–1.020) |
| **Alcohol (SDU), IQR** | 0(0–4.5) | 0(0–3.5) | 0.498 | |
| **Charlson index, IQR** | 0(0–0) | 0(0–0) | 0.204 | |
| **Total number of sexual partners, (IQR)** | 50(15.5–175) | 50(15–112) | 0.479 | 1; (0.999–1.001) |
| **>50 sexual partners during their life, (IQR)** | 168(55.4) | 48(52.3) | 0.68 | |
| **Anal partners during previous 12 m. (IQR)** | 1(1–6) | 2(1–10) | 0.213 | 0,991; (0.976–1.006) |
| **Age at first sexual intercourse, (IQR)** | 18.8(±4.3) | 18.7(± 4.2) | 0.863 | 1.008; (0.944–1.008) |
| **Use of condom,** n (%) | 223(73.6) | 61(67) | 0.590 | 1.267; (0.689–2.329) |
| **Genital/anal warts,** n (%) | 71(23.4) | 21(23.1) | 0.994 | 0.556; (0.271–1,114) |
| **Previous history of:** | | | | |
| Syphilis, n (%) | 80(26.4) | 20(21.9) | 0.395 | |
| Other STI, n (%) | 85(28.1) | 23(25.3) | 0.602 | |
| All STI, n (%) | 131(43.3) | 35(38.5) | 0.419 | |
| **Warts, n (%)** | 104(34.3) | 22(24.2) | **0.069** | 1.948; (0.995–3.811) |
| **Time since HIV diagnose** (months), (IQR) | 24(8–82) | 29(6–111) | 0.05 | 0.995; (0.986–1.004) |
| **CD4 at HIV diagnosis** (cells/uL), (± SD) | 446.9(± 315.7) | 474.1(± 230.8) | 0.475 | |
| **CV at HIV diagnosis** (cells/uL), (± SD) | 5.45±6.05 | 5.26± 5.72 | 0.437 | |
| **CD4 nadir** (cells/uL), (± SD) | 361.9(± 240.9) | 404(± 210–55) | 0.136 | |
| **CD4 nadir < 200 cells/uL, n (%)** | 82 (27.1) | 12(13.2) | **0.006** | **2.363; (1.062–5.256)** |
| **CD4 nadir >500 cells/uL, n (%)** | 72(23.8) | 28(30.8) | 0.181 | |
| **Data of baseline visit (V0)** | | | | |
| **CD4 (cells/uL),** (± SD) | 660.355± 295.8 | 815.621± 826.8 | **0.007** | 1; (0.999–1.001) |
| **CD8 (cells/uL),** (± SD) | 1090.8± 546.9 | 1023.8± 465.8 | **0.294** | |
| **CD4/CD8** | 0.708± 0.396 | 1.01± 1.26 | **0.0001** | 0.634; (0.374–1.074) |
| **VL HIV VL (log)** (± SD) | 3.74± 4.3 | 3.14± 3.95 | **0.058** | |
| **VL HIV <50 cop/mL** | 199(65.7) | 70(76.9) | **0.043** | 1; (1–1) |
| **AIDS stage (A3, B3, C),** n (%) | 84(27.7) | 19(20.9) | 0.193 | 84; (27.7) |
| **ART, n (%)** | 256(84.5) | 85(93.4) | 0.029 | 1.468; (0.333–6.482) |
| **NRTI, n (%)** | 226(74.6) | 80(87.9) | 0.005 | 0.366; (0.117–1.147) |
| **NNRTI, n (%)** | 119(39.3) | 33(36.3) | 0.624 | |
| **IP** | 87(28.7) | 20(21.9) | 0.212 | |
| **INRSTI** | 81(26.7) | 39(42.9) | 0.003 | **0.508; (0.277–0.935)** |
| **Median of months of ART,** (IQR) | 15(4–53) | 18(4–64) | 0.310 | 1.003; (0.993–1.013) |
| **Virological failure**, n (%) | 12(3.9) | 5(5.5) | 0.577 | |

*(Continued)*

**Table 5.** (Continued)

| | With HPV-AR N = 303 | Without HPV-AR N = 91 | Bivariate p* | Multivariate OR**; 95% CI |
|---|---|---|---|---|
| **Infection by LR-HPV genotypes, n (%)** | 228(75.2) | 60(65.9) | 0.079 | **1.842; (1.015–3.344)** |

p*: p-value

OR**: crude odds ratio.

**95% CI: 95% confidence interval.** HIV+MSM, **HIV-positive men that have sex with men;** LTI, **Latent tuberculosis infection;** HCV **hepatitis C virus**; HBV, **hepatitis B virus**; HPV, **Human papillomavirus**; EX-IDU, **ex-injecting drug user**; VL, **viral load.** HR-HPV**: High-risk HPV,** LR-HPV-Br**: Low-risk HPV;** LSIL **(low-grade squamous intraepithelial lesion);** HSIL **(high-grade squamous intraepithelial lesion);** ASC **(uncertain significance cytological abnormalities)**

infection by oncogenic HPV genotypes [26]. Other studies have described the presence of genital warts [27] and a deficient immune system [28] as risk factors for developing neoplasias associated with HR-HPV infection.

Analysis of the histology results revealed that the most frequent low-risk genotypes were 6 in normal mucosa, 11 and 42 in LSIL, and 6 in HSIL and that the most frequent high-risk genotypes were 16 and 45 in in normal mucosa, 16 and 55 in LSIL, and 16, 18, 39, 51, 53, 61, and 68 in HSI. The most prevalent genotype in all types of mucosa was 16, which was significantly associated with anomalous histology (LSIL+). Previous studies in seropositive and seronegative populations [29], in both males and females [30], and in individuals with different sexual orientations [31] found genotype 16 to be the most prevalent virus in anal mucosa and to be associated with the presence of HSIL in patients infected with other oncogenic viruses [32]. Genotypes 18, 53, 61, and 68 were less prevalent but were also associated with the presence of HSIL+. A recent prospective study found that the sole predictive factor for anal HSIL in HIV+ MSM was the presence of oncogenic viruses, including 18 [33].

Besides a previous investigation in our cohort (16), only one study has been published [7] on the prevalence of HPV genotypes in HIV+ MSM according to biopsy results, and it only examined HPV genotypes in individuals with pathological cytology and anoscopy, limiting the information provided [7].

One out of every seven men in our cohort had HSIL (12.3%) or ASCC (0.49%). The HSIL prevalence of HSIL was 21.7%, with an incidence of 30.86/1,000 patient-years, and the prevalence of ASCC was 0.74%, with an incidence of 81.22/100,000 patient-years. Since the first HIV epidemic, the incidence of this neoplasia has increased in seropositive patients, mainly in elderly MSM with AIDS [34]. Recently published data from the Australian HIV and Cancer National Registry showed that the incidence of ASCC rose from 14.8/100,000 people/yr between 1982 and 1995 to 62.1/100,000 people/yr between 2009 and 2012 (p <0.001) [35], similar to the present findings.

Around one-quarter of the HSIL+ mucosa samples had normal cytology; hence, according to current HSIL/ASCC screening recommendations [36], 11 (6.7%) patients (10 with HSIL and 1 with ASCC) would not have been diagnosed or treated. In contrast, no HSIL + samples were found among those with normal cytology and negative HR-HPV PCR. Anal cytology was only moderately sensitive in our patients, who had excellent immune status. This technique may be more sensitive when used in patients with high-degree lesions, those with more affected quadrants affected, and HIV+ patients with CD4 < 200 cells/uL [37]. According to our findings, screening programs for HIV+MSM should include a PCR study of those with normal cytology results to definitively rule out HSIL+. The high frequency of false-negative anal cytology findings in intraepithelial neoplasia screening has led to proposals for the direct performance of HRA [38]; although a recent systematic review called

**A. Genotypes of HPV and normal mucosa**

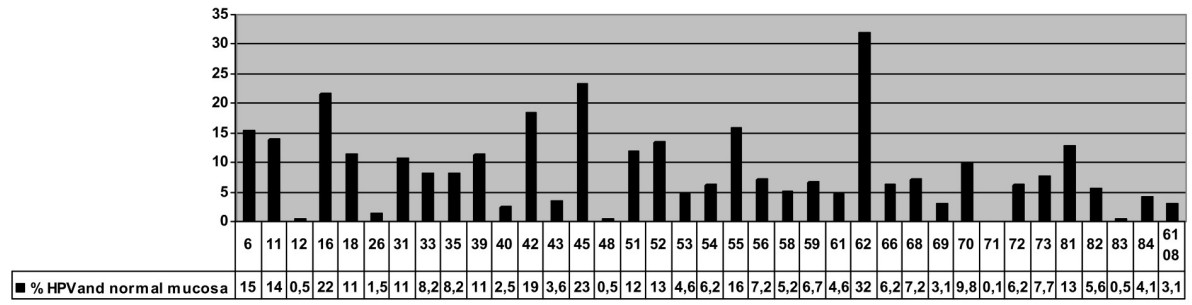

**B. Genotypes of HPV and LSIL.**

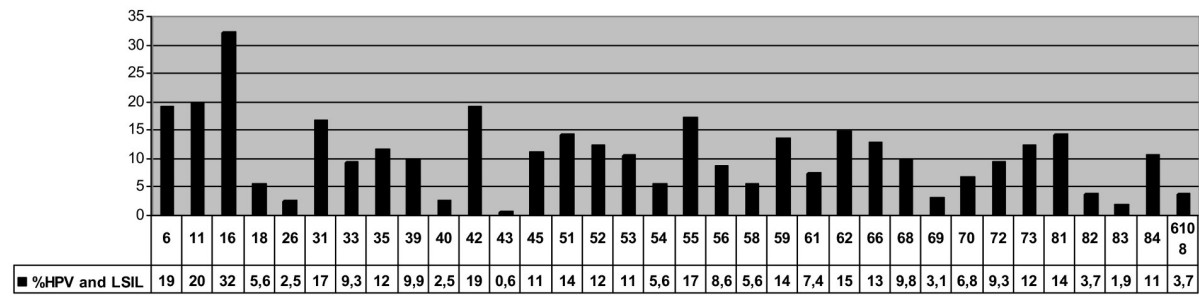

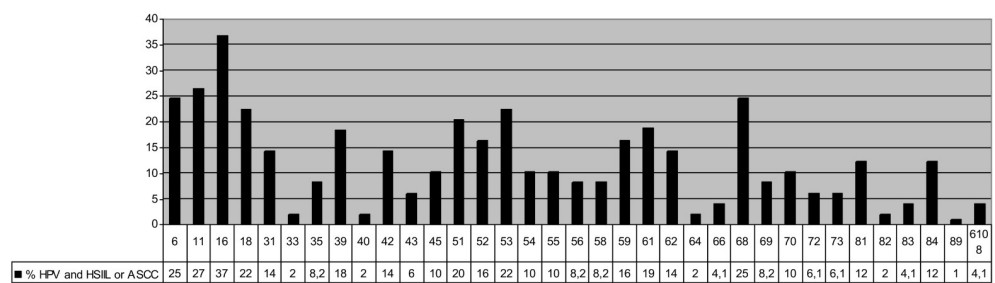

**C. HPV and HSIL+.**

**Fig 2.** A. Genotypes of HPV in normal mucosa. B. Genotypes of HPV in LSIL. C. Genotypes of HPV in HSIL+.

for further research to compare the effectiveness of HRA with that of other techniques in populations at risk [39].

The study was limited to a single center but enrolled a representative sample of a specific group of seropositive patients, and the results may be extrapolatable to similar types of patient. A major study strength is that it reports on the first prospective cohort of HIV+ MSM, demonstrating the value of implementing a program to screen for, diagnose, treat, and follow up HSIL+ lesions.

In conclusion, clearance of oncogenic genotypes in the anal mucosa of HIV+ MSM appears to be positively associated with the duration of ART, although the clearance rate does not affect the incidence of HSIL or ASCC. The reduced incidence of HSIL+ in this population may be attributable to the combined impact of the measures adopted at our center, including screening for lesions and their treatment and follow-up.

**Table 6. Correlation of anal cytology and HPV PCR with histology.**

| | Normal (n = 196) n (%); p* | LSIL (AIN 1) (n = 163) n (%); p* | HSIL (AIN2 and 3) (n = 50) n (%); p* | ASCC (n = 1) n (%); p* |
|---|---|---|---|---|
| **Normal Cytology (n = 165)** | 111 (56.6); 0.0001 | 47 (28.8); 0.0001 | 10 (20); 0.001 | 1 (100); 0.416 |
| **Abnormal Cytology (n = 232)** | 85 (43.4); 0.0001 | 116 (71.2); 0.0001 | 40 (80); 0.001 | 0; 0.416 |
| **LSIL (n = 190)** | 64 (32.7); 0.0001 | 99 (60.7); 0.0001 | 32 (64); 0.015 | 0; 0.337 |
| **HSIL (n = 13)** | 3 (1.5); 0.054 | 3(1.8); 0.18 | 7 (14); 0.0001 | 0; 1 |
| **ASCUS (n = 29)** | 18 (9.2); 0.155 | 14 (8.6); 0.412 | 1(2); 0.153 | 0; 1 |
| | Normal (n = 194) n(%); p* | LSIL (n = 162) n(%); p* | HSIL (n = 49) n(%); p* | SCCA(n = 1) n(%); p* |
| **HPV- High-risk (n = 303)** | 135(69.6); 0.001 | 137(84.5); 0.003 | 42(85.7); 0.118 | 1 (100); 0.58 |
| **HPV-Low-risk (n = 288)** | 136 (70.1); 0.187 | 125(77.2); 0.128 | 38(77.5); 0.45 | 1(100); 0.1 |
| **HPV-High and low-risk (n = 228)** | 101(52); 0.016 | 108(66.7); 0.004 | 31(63.3); 0.435 | 1(100); 1 |
| **HPV-High-Risk negative and normal cytology (n = 48)** | 40 (20.4); 0.0001 | 8 (4.9); 0.0001 | 0; 0.005 | 0; 1 |
| | Normal (n = 190) n(%); p* | LSIL (n = 161) n(%); p* | HSIL (n = 49) n(%); p* | ASCC (n = 1) n(%); p* |
| **HPV- High-risk and normal cytology (n = 114)** | 67(35.3); 0.012 | 39 (24.2); 0.064 | 11 (22.4); 0.259 | 1(100); 0.293 |

p* = p-value

ASCUS: atypical squamous cells of unknown significance; LSIL: low-grade squamous intraepithelial lesions; HSIL: high-grade squamous intraepithelial lesions; AIN: anal intraepithelial neoplasia. ASCC: anal squamous cell carcinoma

## Supporting information

**S1 File. HIV MSM cohort database (VPH-Plos one.sav).**
(SAV)

## Acknowledgments

The authors are grateful to Mercedes Álvarez Romero for her collaboration in coordinating patients, and drawing blood samples; and to Marina Gutiérrez and Rodrigo López of the Pathology Department for processing the samples.

## Author Contributions

**Conceptualization:** Carmen Hidalgo-Tenorio, Mohamed Omar.

**Data curation:** Concepción Gil-Anguita, Mohamed Omar.

**Formal analysis:** Carmen Hidalgo-Tenorio.

**Investigation:** Carmen Hidalgo-Tenorio, Concepción Gil-Anguita, Juan Pasquau.

**Methodology:** Concepción Gil-Anguita, Mohamed Omar, Javier López-Hidalgo, Juan Pasquau.

**Project administration:** Carmen Hidalgo-Tenorio.

**Resources:** Javier López-Hidalgo.

**Software:** Concepción Gil-Anguita.

**Supervision:** Carmen Hidalgo-Tenorio, Miguel Angel López Ruz, Mohamed Omar, Juan Pasquau.

**Validation:** Carmen Hidalgo-Tenorio, Miguel Angel López Ruz, Mohamed Omar, Javier López-Hidalgo, Juan Pasquau.

**Visualization:** Carmen Hidalgo-Tenorio.

**Writing – original draft:** Carmen Hidalgo-Tenorio, Concepción Gil-Anguita.

**Writing – review & editing:** Juan Pasquau.

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
