## [Decision Letter · Decision Letter 0]

16 Sep 2019

PONE-D-19-18194

ART is key to clearing oncogenic HPV genotypes (HR-HPV) in anal mucosa of HIV-positive MSM

PLOS ONE

Dear Dr Tenorio,

Thank you for submitting your manuscript to PLOS ONE. After careful consideration, we feel that it has merit but does not fully meet PLOS ONE’s publication criteria as it currently stands. Therefore, we invite you to submit a revised version of the manuscript that addresses the points raised during the review process.

In agreement with the reviewer, it appears that your manuscript is of interest regarding the impact of HPV anal infection in MSM . However it is clearly in need of clarifications with respect to the HPV genotypes, and the histological analyses (figures 1 & 2).

The section on anal cytology and HPV genotyping by PCR (comment 6) is very hard to follow, and hence conclusions could be mistaken. This important section really needs clarification.

See other comments by the reviewer.

We would appreciate receiving your revised manuscript by Oct 31 2019 11:59PM. To enhance the reproducibility of your results, we recommend that if applicable you deposit your laboratory protocols in protocols.io, where a protocol can be assigned its own identifier (DOI) such that it can be cited independently in the future. For instructions see: http://journals.plos.org/plosone/s/submission-guidelines#loc-laboratory-protocols

We look forward to receiving your revised manuscript.

Kind regards,

Jean-Luc EPH Darlix, MG, Ph.D.

Academic Editor

PLOS ONE

Journal Requirements:

1. Thank you for including your funding statement;"The funders had no role in study design, data collection and analysis, decision to publish, or preparation of the manuscript."

Please provide an amended Funding Statement that declares *all* the funding or sources of support received during this specific study (whether external or internal to your organization) as detailed online in our guide for authors at http://journals.plos.org/plosone/s/submit-now.  

Please state what role the funders took in the study.  If any authors received a salary from any of your funders, please state which authors and which funder. If the funders had no role, please state: "The funders had no role in study design, data collection and analysis, decision to publish, or preparation of the manuscript."

Reviewers' comments:

Reviewer's Responses to Questions

**Comments to the Author**

1. Is the manuscript technically sound, and do the data support the conclusions?

Reviewer #1: Yes

2. Has the statistical analysis been performed appropriately and rigorously? 

Reviewer #1: Yes

3. Have the authors made all data underlying the findings in their manuscript fully available?

Reviewer #1: No

4. Is the manuscript presented in an intelligible fashion and written in standard English?

Reviewer #1: Yes

5. Review Comments to the Author

Reviewer #1: General

This is a very important study exploring the natural history of anal HPV infection and anal dysplasia among HIV+ positive MSM. The study was well executed and has robust analysis. It also explored treatment of HSILs with a variety of methods including impact of giving the quadrivalent vaccine. I recommend the manuscript for publication after the comments highlighted below are addressed.

Methods:

Comment 1: Design: 5 paragraph last sentence:

“At the same visit, biopsies were taken using an endoscopic retrograde cholangiopancreatography catheter.”

What biopsies were these? Anal or liver?

Comment 2: Definition of variables

“Histology with anal HSIL+: from HSIL to ASCC.”

“Histology with LSIL+: from LSIL to ASCC.”

These are not clear and need to be updated.

Results

Comment 3: 5. HPV genotype distribution by histology

Text in this section and reference to the tables needs to be clarified.

For example the second paragraph “As depicted in Figure 1, none of the 36 HPV genotypes isolated in normal mucosa were significantly associated with this histological finding (normal mucosa), and the most frequently isolated genotypes were 6 (15.4%), 16 (21.6%), 42 (18.5%), 45 (23.2%), and 62 (31.9%).”

Comment 4: The information on the most common genotypes in actually in Figure 2 not figure 1 which shows the stabilisation of the HSIL+ rates

Comment 5: Again the third paragraph, “In LSIL+ mucosa samples, the most frequently isolated genotypes were 6 (19.1%), 11 (19.8%), 16 (32.3%), 31 (16.7%), 42 (19.1%), and 55 (17.3%), as shown in Figure 2. Genotype 66 was significantly associated with LSIL (12.9% vs. 5.6% p=0.001) (Figure 1).”

It is not immediately clear from Figure 1 that type 66 is associated with LSIL

Comment 6: 6. Correlation of anal cytology and HPV PCR with histological findings

“The cytology study showed that 165 samples were normal and 232 had some degree of

dysplasia. Among the patients with dysplastic samples, 36.6 % had normal anoscopy

(p=0.0001), 50% LSIL (p=0.0001), and 17.2% HSIL (p=0.001).” This section is difficult to follow for the readers. For example you would need to calculate that (85/232)=36.6 had normal. There is another % i.e..43.4% which is next to 85 and can be confusing to the reader.

Comment 7: Table 1: As with other abbreviations, can the authors put the full name for the SDU in table footer

Comment 8: Table 5: Needs a bit of formatting so that all the text and number are of the same font.

Discussion

Comment 9: Second paragraph. This sentence is missing the 95% CI “The duration of ART was the sole factor favoring oncogenic virus clearance (OR 1.016; 95% CI),”

Comment 10: Third paragraph. It is this sentence which explains what Figure 1 showed “However, we have observed a significant reduction in the HSIL+ rate among HIV+ MSM since the start of the program in 2010, with a stabilization since 2016”

However, this is never mentioned in the results section. It then ties with observation and comments in the results section that the part 5 and 6 of the results needs to be revised as they are rather confusing to the reader.

Comment 11: Fourth paragraph. The finding on nadir CD4+ is important “Factors related to infection with HPV oncogenic virus were a history of CD4 nadir < 200 cells/uL (OR: 2.363, 95% CI: 1.062-5.256), and simultaneous infection by low- and high risk HPV genotypes (OR: 1.842; 95% CI: 1.015-3.344).”

Can the authors mention that this confirms previous findings and cite some few references

Comment 12: Limitations paragraph: The spelling for HIV needs to be updated “A major study strength is that it reports on the first prospective cohort of VIH+ MSM, demonstrating the value of implementing a program to screen for, diagnose, treat, and follow up HSIL+ lesions.”

6. PLOS authors have the option to publish the peer review history of their article (what does this mean?). If published, this will include your full peer review and any attached files.

Reviewer #1: Yes: Dr Admire Chikandiwa

---

## [Author Response · Author response to Decision Letter 0]

2 Oct 2019

We are grateful to the reviewers for their valuable comments, which have all been taken into account in our revision, as explained in our point-by-point responses.: 

Reviewer #1: General

This is a very important study exploring the natural history of anal HPV infection and anal dysplasia among HIV+ positive MSM. The study was well executed and has robust analysis. It also explored treatment of HSILs with a variety of methods including impact of giving the quadrivalent vaccine. I recommend the manuscript for publication after the comments highlighted below are addressed.

Methods:

1- Comment 1: Design: 5 paragraph last sentence:

“At the same visit, biopsies were taken using an endoscopic retrograde cholangiopancreatography catheter.”

What biopsies were these? Anal or liver?

We have now specified in the revised text that we took biopsies of the anal mucosa, as follows:

“At the same visit, biopsies of the anal mucosa were taken using an endoscopic retrograde cholangiopancreatography catheter”. 

2- Comment 2: Definition of variables

“Histology with anal HSIL+: from HSIL to ASCC.”

“Histology with LSIL+: from LSIL to ASCC.”

These are not clear and need to be updated.

We have clarified these definitions. 

Results

3- Comment 3: 5. HPV genotype distribution by histology

Text in this section and reference to the tables needs to be clarified.

For example the second paragraph “As depicted in Figure 1, none of the 36 HPV genotypes isolated in normal mucosa were significantly associated with this histological finding (normal mucosa), and the most frequently isolated genotypes were 6 (15.4%), 16 (21.6%), 42 (18.5%), 45 (23.2%), and 62 (31.9%).”

We have clarified the text and reference accordingly in the revised Results section.

4-Comment 4: The information on the most common genotypes is actually in Figure 2 not figure 1 which shows the stabilisation of the HSIL+ rates. 

We apologize for this typographic error, which has been amended. 

5- Comment 5: Again the third paragraph, “In LSIL+ mucosa samples, the most frequently isolated genotypes were 6 (19.1%), 11 (19.8%), 16 (32.3%), 31 (16.7%), 42 (19.1%), and 55 (17.3%), as shown in Figure 2. Genotype 66 was significantly associated with LSIL (12.9% vs. 5.6% p=0.001) (Figure 1).”

It is not immediately clear from Figure 1 that type 66 is associated with LSIL

 We are again grateful to the reviewer for detecting these errors. These sentences now read as follows: 

In LSIL+ mucosa samples, the most frequently isolated genotypes were 6 (19.1%), 11 (19.8%), 16 (32.3%), 31 (16.7%), 42 (19.1%), and 55 (17.3%), as shown in Figure 2B. Genotype 66 was significantly associated with LSIL (12.9% vs. 5.6% p=0.001). 

6- Comment 6: 6. Correlation of anal cytology and HPV PCR with histological findings

“The cytology study showed that 165 samples were normal and 232 had some degree of dysplasia. Among the patients with dysplastic samples, 36.6% had normal anoscopy (p=0.0001), 50% LSIL (p=0.0001), and 17.2% HSIL (p=0.001).” This section is difficult to follow for the readers. For example you would need to calculate that (85/232)=36.6 had normal. There is another % i.e..43.4% which is next to 85 and can be confusing to the reader.

We have clarified the text accordingly, making these results easier to follow.

7- Comment 7: Table 1: As with other abbreviations, can the authors put the full name for the SDU in table footer.

This has been done.

8- Comment 8: Table 5: Needs a bit of formatting so that all the text and number are of the same font.

This has been done. 

Discussion

9- Comment 9: Second paragraph. This sentence is missing the 95% CI “The duration of ART was the sole factor favoring oncogenic virus clearance (OR 1.016; 95% CI),”

This typographic error has now been amended. 

10- Comment 10: Third paragraph. It is this sentence which explains what Figure 1 showed “However, we have observed a significant reduction in the HSIL+ rate among HIV+ MSM since the start of the program in 2010, with a stabilization since 2016”

However, this is never mentioned in the results section. It then ties with observation and comments in the results section that the part 5 and 6 of the results needs to be revised as they are rather confusing to the reader.

- This information is given in the Part 2 of Results under the heading “Clearance and acquisition of HR-HPV infection and prevalence of HSIL+ in anal mucosa….”: However, as depicted in Figure 1, we found significant reductions in HSIL+ cases between 2010 and 2018 (42.9% vs. 4.1%, p=0.034), between 2010 and 2013 (42.9% vs. 13.8%, p=0.003), and between 2013 and 2016 (13.8% vs. 4.8%, p=0.0001), followed by a stabilization between 2016 and 2018 (4.8% vs. 4.1%, p=0.617) (Figure 1). …

We have clarified parts 5 and 6 of Results, as requested.

11- Comment 11: Fourth paragraph. The finding on nadir CD4+ is important “Factors related to infection with HPV oncogenic virus were a history of CD4 nadir < 200 cells/uL (OR: 2.363, 95% CI: 1.062-5.256), and simultaneous infection by low- and high risk HPV genotypes (OR: 1.842; 95% CI: 1.015-3.344).”

Can the authors mention that this confirms previous findings and cite some few references

This is done in the following paragraph: 

“These data confirm recent observations of the protective effects ofART and CD4 levels>500 cells/uL against infection by oncogenic HPV genotypes (26). Other studies have described the presence of genital warts (27) and a deficient immune system (28) as risk factors for developing neoplasias associated with HR-HPV infection”. 

12- Comment 12: Limitations paragraph: The spelling for HIV needs to be updated “A major study strength is that it reports on the first prospective cohort of VIH+ MSM, demonstrating the value of implementing a program to screen for, diagnose, treat, and follow up HSIL+ lesions.”

This has been changed to “HIV+ MSM”

With Best regards

Carmen Hidalgo Tenorio

---

## [Editor Report · Decision Letter 1]

8 Oct 2019

ART is key to clearing oncogenic HPV genotypes (HR-HPV) in anal mucosa of HIV-positive MSM

PONE-D-19-18194R1

Dear Dr. Tenorio,

We are pleased to inform you that your manuscript has been judged scientifically suitable for publication and will be formally accepted for publication once it complies with all outstanding technical requirements. Nonetheless your manuscript requires a careful editing at this stage.

With kind regards,

Jean-Luc EPH Darlix, MG, Ph.D.

Academic Editor

PLOS ONE
---

## [Editor Report · Acceptance letter]

14 Oct 2019

PONE-D-19-18194R1 

ART is key to clearing oncogenic HPV genotypes (HR-HPV) in anal mucosa of HIV-positive MSM 

Dear Dr. Hidalgo-Tenorio:

I am pleased to inform you that your manuscript has been deemed suitable for publication in PLOS ONE. Congratulations! Your manuscript is now with our production department. 

With kind regards,

on behalf of

Professor Jean-Luc EPH Darlix 

Academic Editor

PLOS ONE